# Rwanda's Land Policy Reform: Self-Employment Perspectives from a Case Study of Kimonyi Sector

**Mireille Mizero [1,\*], Aristide Maniriho [1,2]**  **, Bosco Bashangwa Mpozi [3], Antoine Karangwa [4], Philippe Burny [1] and Philippe Lebailly [1]**

1 Department of Economy and Rural Development, Faculty of Gembloux Agro-Bio Tech, University of Liège, Passage des Déportés 2, 5030 Gembloux, Belgium; aristide.maniriho@uliege.be (A.M.); philippe.burny@uliege.be (P.B.); philippe.lebailly@uliege.be (P.L.)

2 Gikondo Campus, College of Business and Economics, University of Rwanda, KK 737 Street, Kigali PO BOX 4285, Rwanda

3 Higher Institute of Development Techniques of Mulungu, Bukavu, Democratic Republic of the Congo; mpoziba@yahoo.fr

4 School of Agriculture & Food Sciences, College of Agriculture and Veterinary Medicine, University of Rwanda, Musanze District, Busogo PO BOX 210, Rwanda; a.karangwa@ur.ac.rw

\* Correspondence: mireille.mizero@doct.uliege.be

**Abstract:** Rwanda's Land Policy Reform promotes agri-business and encourages self-employment. This paper aims to analyze the situation from a self-employment perspective when dealing with expropriation risk in rural areas. In this study, we conducted a structured survey addressed to 63 domestic units, complemented by focus groups of 47 participants from Kimonyi Sector. The binary logistic regression analysis revealed that having job alternatives, men heading domestic units, literacy skills in English, and owning land lease certificates ($p < 0.05$) are positively and significantly related to awareness of land expropriation risk. The decision of the head of the domestic unit to practice the main activity under self-employment status is positively influenced by owning a land lease certificate, number of plots, and French skills, while skills in English and a domestic unit's size have a positive and significant influence on involvement in a second activity as self-employed. Information on expropriation risk has no significant effect on self-employment. The domestic unit survey revealed that 34.9% of the heads of domestic units only have one job, 47.6% have at least two jobs in their everyday life, 12.7% have a minimum of three jobs, and 4.8% are inactive. The focus group synthesis exposed the limits to self-employment ability and facilities.

**Keywords:** land reform; self-employment; own account business; rural development; off-farm jobs; literacy skills; Rwanda

## 1. Introduction

Rwanda's economic reform associated with its market-led model of economy refers to structural transformation, mostly away from subsistence agriculture and towards poverty reduction [1]. The expropriation risk for public interest is a recurrent concern of the urban population in Kigali, a city of Rwanda [2,3]. However, the phenomenon is also present in rural areas. The main reason for expropriation risk is different in rural and urban areas. It is mainly observed at sites where industrial projects and big investors are more privileged than rural land users, not in terms of public interest, but for business purposes. The resilience strategies employed to deal with this expropriation risk still receive less attention in rural areas. Several studies have argued that the Rwandan Government is banking on the capturing of land taxes via deliberate land-based real estate development [4]. The Rwanda post-genocide land policy is a hybrid model designed by both the state and the market-led economy [5,6]. Regardless of the labels and criticisms associated with the Rwandan economic model [4,7,8], rural people do not passively wait for offshore solutions.

Domestic units in rural areas which are facing the constraints of a context of renovation try to implement multiple pathways as job alternatives. A Vietnamese study concluded that the rural population continues to choose both on-farm and off-farm jobs to overcome the constraints of socio-economic changes [9]. Off-farm or on-farm job alternatives do not automatically represent self-employment. This study aims to examine the situation of self-employment in rural areas since the Rwandan Government has reformed land policy and launched the maxim of "*own account jobs creation*", translated in Kinyarwanda language as "*Kwihangira imirimo*". There is no evidence that all land users are advised that the Rwandan Government is the supreme decision maker on land use, despite the systematic registration of land holdings (article 3 of the Law n° 03/2013OL of 16/06/2013 repealing Organic Law n° 08/2005 of 14/07/ determining the use and management of land in Rwanda) [10].

Our research question addresses how Rwanda's rural population is managing the self-employment advice in dealing with land policy reform constraints, especially the land expropriation risk. The assumption is that the information on expropriation risk has engaged rural domestic units in a dynamic range of job alternatives (DJA), rather than self-employment. The significant variables related to the awareness of the land expropriation risk can help us to describe the profile of the relevant population by their professional adaptation in a changing context of land reform. The low risk of expropriation is supposed to allow farmers to liberate capital for other agricultural investments, rather than spending their private resources on protecting their land [11]. On the contrary, the awareness of the eventual risk of land expropriation warns farmers to search for job alternatives as self and/or non-self-employed workers. The methodology applied in this study consisted of an in-depth literature review and field work, in an attempt to try to understand what is happening in Rwanda's rural areas regarding the self-employment situation in relation to land policy constraints. The main constraint is the risk of expropriation due to "irrational land use". The concept of "rational" land use is somehow ambiguous. The rational and irrational land use are described in the Section 2 of Land Obligations in the Law n° 03/2013OL of 16/06/2013 repealing Organic Law n° 08/2005 of 14/07/2005 determining the use and management of land in Rwanda. The "rational" use is appreciated by the authorities referred to articles number 39, 40 and 41. The significance of rational land use in the French version is equivalent to "sustainable" land use in the English version of the cited Organic Law. The "modalities of protecting and sustainable use of land shall be determined by an order of the Prime Minister" (article 28). To limit the land expropriation risk for irrational use and increase production in a collective approach to land management, small farmers are engaged in a broad program of land consolidation. Nevertheless, investigating the limits of land consolidation politics is not the purpose of this paper. For more detail, several studies are dedicated to evidence of land consolidation, agricultural specialization, and the impact on food security [12,13].

The promotion of non-farm activities has been recommended due to the risk of a "large number of people who may become landless in the near future" [14]. The recommendation for off-farm job improvement is very popular in the context of land scarcity and long-lasting land conflicts. However, studies which have examined the problem of self-employment related to land reform are rare. The changes in land policy occurred in a specific context of national reconstruction in the aftermath of the war and genocide against the Tutsi (1990–1994) [14]. Furthermore, Rwanda is a renovating agricultural society where the number of big land deals is increasing, but not necessarily for agricultural purposes [15]. The consequence of land policy change is that the population depending on rural land for their livelihood may need to deal with its professional conversion during and after reforms, without being able to significantly strengthen their skills before land policy implementation. One of the challenges of the state- and market-led land policy reforms is how the population can adjust their professional profile and skills. Self-job diversification is considered to be a gauge of awareness of current and further agrarian and land use changes. The Government is encouraging the population to achieve their own account business. Our paper assesses

the job alternatives in rural areas where agriculture is the main occupation of domestic units. It is focused on the rural population facing the eventual risk of expropriation.

The limit of our research is that it is only oriented towards self-employment in rural areas, while land policy is global and expropriation risk is highly likely to be observed in urban areas. Land policy has considered rural and urban land use and management separately [16]. Dissimilarities between rural and urban settings could influence the options for professional conversion of the rural people when they are looking for new geo-localized job alternatives. The virtual separation of urban areas is due to the fact that job opportunities are dissimilar in these contrasting, but complementary, environments. There is a high risk of impeding a holistic approach by creating artificial boundaries [17]. On the one hand, off-farm employment is supposed to be abundant in urban areas and the surrounding zone, even if job-seekers live in or come from rural areas, especially in the agricultural sector [9]. On the other hand, rural areas are continuously being converted into urban zones as the Rwandan "master plan" of land use is supposed to be updated every five years. Therefore, the new shape of land boundaries could impact the situation of individual and collective job perspectives in rural areas. Section 2 presents the cross-country situation of self-employment globally. Section 3 briefly talks about labor force transformation and the disproportional balance of remuneration. Section 4 describes the study area, method of sampling, data collection, and analysis. The findings are presented and discussed, respectively, in Sections 5 and 6. Section 7 includes the paper's conclusion and policy perspectives.

## 2. Self-Employment Situation in the World: Cross-Country Disparities

According to the World Bank (WB) and International Labour Organisation (ILO), rural and agricultural societies have high levels of self-employed workers compared to industrial and non-rural societies. Figure 1 illustrates the differences between five countries of the East Africa Community (Burundi, Rwanda, Tanzania, Uganda, and Kenya), Sub-Saharan Africa (SSA), excluding high-income countries such as South Africa; four countries from industrialized regions (Canada, Belgium, and Germany); and the European Union. The situation of the first group is critical, with more than a 70% self-employed population, while in the second group the self-employed population represents about 20%. Kenya's self-employed population has a lower than average value for SSA and has gradually decreased since 2006 compared to other countries of the East Africa Community. Each country has its own working status scheme, even though the ILO has internationally recommended standardized norms [18]. Rwanda's self-employment situation was in the same range as that of other countries of the East Africa Community from 1991 until 2010. However, trends in self-employment in terms of the total percentage of employment assessed annually by the WB and modeled by ILO estimation are gradually decreasing. The main change occurred at the beginning of 2014, when the definition of one's employment status was adapted to the concept adopted by the International Labour Office [19]. A high ratio of self-employed workers is not a positive indicator of economic performance:

> " . . . *A high proportion of wage and salaried workers in a country can signify advanced economic development. If the proportion of own-account workers (self-employed without hired employees) is sizeable, it may be an indication of a large agriculture sector and low growth in the formal economy*" [18].

The definition of working status and data collection are not the same from one country to another. This is why the comparison is biased. For example, women's activities could be considered both economic and non-economic regarding social norms, cultural trends, and the legal framework. This constitutes a limitation on the comparability of global data [18]:

> "*Self-employed workers are those workers who, working on their own account or with one or a few partners or in cooperative*" [18].

Rwanda's Government is engaged in a broad program of labor force transformation [20]. The conversion of workers theoretically follows the remuneration gradient.

However, the situation is more complex than the evidence suggests. Agriculture remains the main source of employment but is a less remunerated sector compared to other socioeconomic activities, as detailed in the following section.

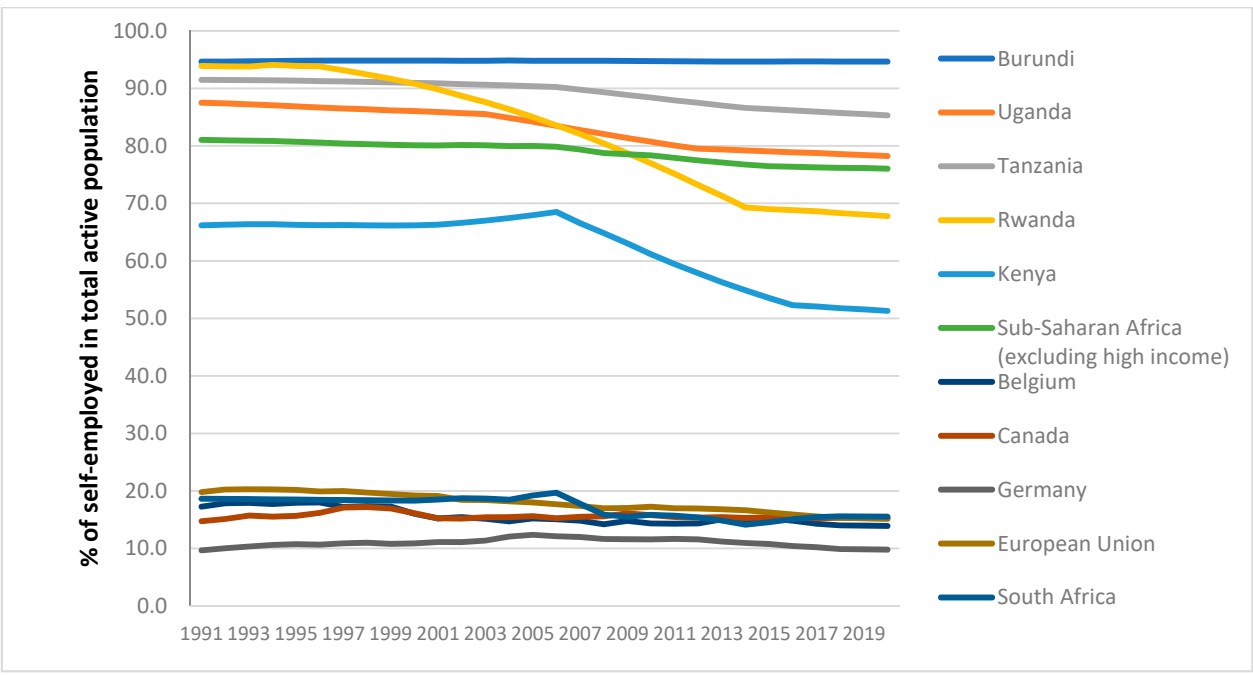

**Figure 1.** Cross-country self-employment in several industrial and non-industrial countries. Source: Authors from Open Data of the World Bank Group [18].

## 3. Rwanda's Labor Force Transformation and Remuneration Disparities

Rwanda Vision 2020 aims to make Rwanda a middle-income country through an extensive program of transforming the labor force according to the requirement of the labor market [1]. For this perspective, one of the goals of Vision 2020 is the creation of *"200,000 off-farm jobs annually to speed the development growth"* [20]. The percentage of the employed population in the market oriented towards agriculture and the supposed market-led land reforms was estimated to be 37.7% in August 2017 and the average number of off-farm jobs in the same period was 1,799,080 for both primary and secondary jobs [20].

The labor force survey revised its methodology and indicators in 2017, according to the new international standards of employment analysis [20]:

*"Employment includes only persons working for pay or profit, excluding persons engaged wholly or mostly in subsistence foodstuff production. The effect of this is to lower the count of employment (according to the old definition) and to higher the count of unemployment because some of the subsistence foodstuff producers would be looking and available for paid work for profit and thus be classified as unemployed"* [20].

The agricultural sector in Rwanda involves less profitable jobs per hour in comparison to other economic sectors of activities, according to the National Labor Force Survey. An agricultural worker receives 33% of the hourly compensation provided in the services sector, 52% compared to the average hourly wage in the country, and 47% compared to industrial employment. In agriculture, the average daily wage is below the poverty line of US $0.8, compared to $2.6 per day in industry and $4.2 per day in services [20]. In spite of such disparities in the remuneration between the on-farm and off-farm economy, the two sectors are positively linked, as opportunities allow for compensation for the low incomes in the agricultural sector. The National Institute of Statistics of Rwanda (NISR) has reported that the agricultural population is decreasing in size.

## 4. Material and Method

### 4.1. Study Area

The field work was carried out in a small but representative region. Northern Rwanda is known for having the highest agricultural potential, being considered the "*Silo of the country*". The survey was carried out in four villages of Kimonyi Sector, Musanze District. Kimonyi Sector is one of the 15 Sectors of the District of Musanze (Figure 2). This Sector is entirely located in a rural area. Its landscape is dominated by volcano rocks rich in cement raw material. An important cement factory (600,000 tons per year) and construction businesses are being implemented in the industrial bloc of Kimonyi Sector in Musanze District. The factory has supposedly created 600 jobs and improved construction material in terms of both quality and quantity [21,22]. The land use of the site has changed from agricultural activities to industrial cement production. The former agricultural operators were expropriated with compensation and went to search for other land in order to maintain their activities or find other job opportunities.

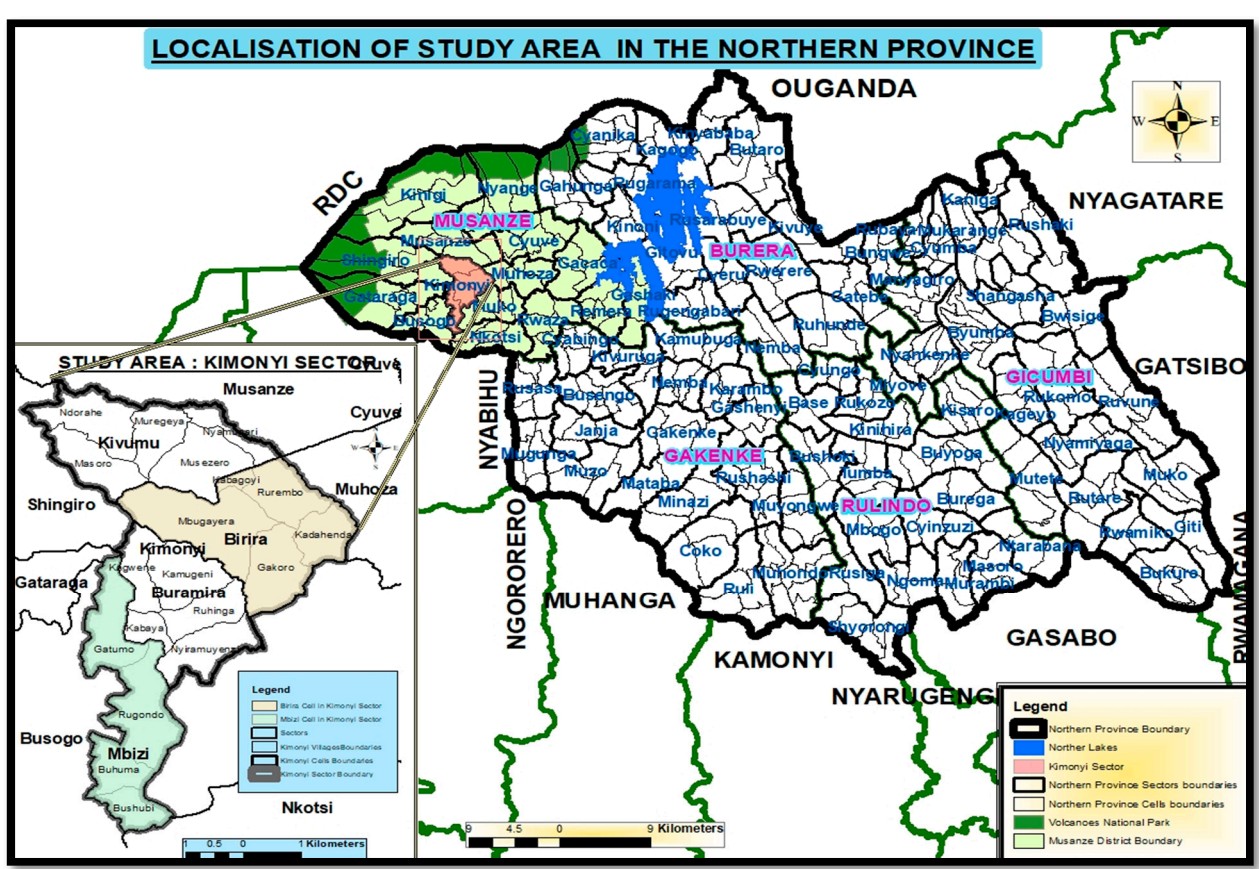

**Figure 2.** Study area of Kimonyi Sector, Musanze District, in the Northern Province. Source: One Stop Centre, Musanze District.

### 4.2. Level of the Structured Interviews: Domestic Unit

For the in-depth interviews, the "domestic unit" terminology was adopted, instead of the classic "household", because traditional households in the post-genocide era have a particular status. The paradigm of the domestic unit has impacts on how the members are managing their land rights in the transition from being customers during the current land tenure reform, especially for gender mainstreaming [23–26]. Relatives living far away from the domestic units were not considered in the sampling method because they were not sharing their everyday life with the domestic unit during the survey [27]. However, if they are mentioned on the land lease certificate, they must be consulted about any decision on land use or allocation. The head of the domestic unit (man or woman) is the person who

is mentioned at the top of the list of co-proprietors. When the domestic unit does not yet have a formal registered land lease certificate, the head of the domestic unit is the person who is supposed to stand for others in the process of land registration and in any kind of collective and recognized mandate of representation. The concept is close to the philosophy of indigenous common land tenure in many countries of Africa, where individualistic land ownership is exceptional [28]. The Rwandan land reform validated the traditional status of common land-ownership, while recognizing individual rights to shared land by registering all share-holders on the same document of the land lease certificate.

### 4.3. Survey Mapping and Sampling Method

Four villages (Gakoro, Mbugayera, Kabagoyi, and Rurembo) were sampled for structured and semi-structured interviews. Two focus groups were organized in two cells. The first was in Birira Cell and the second in Mbizi Cell. The field work began on 20 July and lasted until 12 September 2018. The non-random sample was established based on voluntary participation in the structured interviews or focus groups. The role of the administrative authorities at the cell level was limited to easing contact with the population.

The sample size was determined by using the formula below [29]:

$$n = \frac{N}{1 + N(e)^2}$$

where:

*n* is the sample size,

*N* is the resident population size in Kimonyi Sector based on the Fourth National Census of Population and Housing (14,107 inhabitants) [30], and

*e* is the 10% error assumed. This large margin of error assumed in the sampling method is due to the methodology adopted, which included a small sample in a small region [27].

With respect to the sampling formula, the size of the sample was equal to 110 domestic units. The sample size was separated into two groups of 60% and 40%. The first group provided answers to a structured questionnaire and the second was involved in semi-structured interviews. Such a field work strategy is a holistic approach to rapidly appreciating the complexity of the situation regarding self-job creativity. The 60–40 method allows a wide range of information to be collected in a relatively short period of time and smart field work constraint management. Rural housing in Rwanda is still dispersed, despite the villagization policy. The access paths to domestic units are not very practical because of the mountainous landscape. The first step of the survey was conducted with a structured questionnaire addressed to 63 heads of domestic units in four villages of Birira Cell, Kimonyi Sector. The second part (47 domestic units) involved semi-structured interviews with focus groups. The first part of the survey was treated with quantitative analysis and the second one with qualitative analysis. The 60–40 strategy of field data collection constitutes an equilibrium feature of both the structured survey and the focus group's large debate. In addition, the method allows to extend interviews outside of the research area, in order to capture other cross-referenced information. We took advantage of the "land week" organized in the neighboring District of Rulindo to attend a large public event, where we observed how express land registration services were delivered to the Northern Province population. There, we met different job makers and saw many processed agricultural products. The event gave us the impression that the agro-industry sector and connected services could represent an important labor force in the study area.

### 4.4. Data Analysis and Variables

During the field work, we noticed that some respondents were unaware that their land certificates were not definitely private property titles. They also ignored the fact that the land they use can change at any time, according to the will of the Government. Therefore, we proceeded in four steps. We decided to first focus on the identification of significant indicators related to the awareness of a possible situation of expropriation with

or without compensation. Then, we identified the activities performed by active members of domestic units created by themselves or other employees. The methods allowed us to observe that not all practiced jobs were created by workers as self-employment. We were also interested in recording the number of combined activities pursued by the heads of the domestic units. For the binary logistic regression analysis, we considered 12 independent variables to analyse the land user's information on land expropriation risk ($Y$) as a dependent variable. The literacy skills refer to different aptitudes of the heads of domestic units ($X_1$ = reading Kinyarwanda, $X_2$ = reading French, $X_3$ = reading English, $X_4$ = reading Swahili, and $X_5$ = reading other languages). These aptitudes are required in numerous communications regarding public or private job recruitment or for one's own account business planning and management. Participation in agricultural activities ($X_6$) was considered an indicator of heads of domestic units being informed about their obligation to add value to land holdings in order to prevent the risk of expropriation for non-rational use. Land lease certificate holdings ($X_7$) were considered an indicator of awareness of the risk of non-recognized land rights. The rational use of land ($X_8$) was addressed by an indirect question on the ability of domestic units to achieve management of their entire land holdings. The education level ($X_9$) was taken into account to check whether the level of education influenced information accessibility regarding the process of land expropriation. The number of jobs recorded ($X_{10}$) corresponds to economic activities performed by the heads of domestic units. This indicator allows us to check if this is a coping strategy related to information on the risk of expropriation. The sex ($X_{11}$) was also considered because the current land reform policy recognizes equal rights for both men and women in relation to land holdings. Male- and female-headed domestic units are supposed to have the same chance of accessing information on the expropriation process. The age of the heads of domestic units ($X_{12}$) was considered as this can limit the ability to get or create one's own job if one's land rights are restricted or lost. The Binary Logistic Regression model [31] is presented below (Equation (1)). The Backward Selection method in R was used to determine the most reliable model via which significant variables are influencing the access to information about land expropriation risk. The lowest value of the Akaike Information Criterion (AIC) determined the most suitable indicators in the model.

$$Y = \alpha + \sum_{k=1}^{n} \beta k X k + \varepsilon, \; n = 12, \tag{1}$$

$$AIC = 2k - 2\ln(L), \tag{2}$$

where $k$ is the number of estimated parameters and L is the maximum of the likelihood function.

As the specified model is not linear (Equation (1)) but binary (since the dependent variable Y takes only two values, 1 or 0), we used the maximum likelihood method to estimate the coefficients $\alpha$ and $\beta k$. The results are presented and discussed in the following sections.

The binary logit also estimates factors affecting the decision of the domestic unit to work under self-employed status for the principal or secondary activity. The analysis takes 13 variables into account.

## 5. Results

More than 50% of heads of the domestic units are not informed about the expropriation risk (Table 1). They ignore the terms ending the land use contract when the authorities revise the "master plan". They believe that having a land lease certificate guarantees ownership of the land. For instance, the interviews showed that the content of land certificates has not been well-understood. The number of years of validity of the land lease certificate is clearly mentioned. After these terms, land users must request renewal of land lease certificates. The revision of the "master plan" occurs at least once every five years. This is likely to affect the land allocation previously decided upon for the land holdings

located at the concerned site. The advantage for those who at least have at a land certificate is that they receive prior compensation corresponding to the registered area per square meter. Other users without land lease certificates could lose their land rights without compensation. The lack of information on this issue is a problem because it may lead to unpleasant surprises. The anticipation of planning for long-term land use changes becomes problematic for misinformed land users.

**Table 1.** Proportion of domestic units informed of the expropriation risk in Kimonyi Sector.

| Information on Expropriation Risk | Frequency | Percentage |
|---|---|---|
| The head of the domestic unit is not aware of the expropriation risk in the case of land use change | 35 | 55.6 |
| The head of the domestic unit is aware that they could be expropriated in the case of land use change | 28 | 44.4 |
| Total | 63 | 100.0 |

### 5.1. Indicators of Awareness for Possible Loss of Land Rights

We analysed the indicators related to having information on the prospect of land expropriation. The logistic regression in R (Table 2) shows that job alternatives for the head of the domestic unit, male-headed domestic units, literacy skills in English, and land lease certificate holdings positively and significantly influence awareness of land right restrictions. The most significant indicators in the model were determined with backward selection, with the lowest value of AIC (66.5). The less robust models with a high value for AIC were progressively removed by the software. As the "land rush" is a game of negotiation on land rights, the well-informed, better connected, and better equipped, either financially or in terms of skills, are well-equipped in land deals [8].

**Table 2.** Significant variables related to the land user's information about land expropriation.

| Informed on Expropriation Risk | Coefficient Estimate | Std. Error | z Value | Pr (>\|z\|) |
|---|---|---|---|---|
| (Intercept) | −5.44 | 1.88 | −2.90 | 0.004 ** |
| Reading skills in Kinyarwanda | −1.47 | 0.86 | −1.70 | 0.089 |
| Reading skills in English | 3.53 | 1.44 | 2.44 | 0.015 * |
| Participation in agricultural activities | −1.70 | 0.97 | −1.75 | 0.079 |
| Land lease certificate possession | 1.50 | 0.74 | 2.04 | 0.042 * |
| Job alternatives for the head of the domestic unit | 1.66 | 0.67 | 2.46 | 0.014 * |
| Male−headed domestic units | 3.67 | 1.47 | 2.49 | 0.013 * |

Signif. codes: 0.001 '**' 0.01 '*'.

### 5.2. Job Alternatives for a Domestic Unit's Members in Kimonyi Sector

The important information in this study was the recording of the availability of worker(s) involved in a given job. Even if sporadic or considered minor in the social mind-set, it was recorded as a job alternative existing in the labor market. Table 3 shows the main jobs of active members of the surveyed domestic units. The recorded jobs are a mixture of self-created jobs and other jobs not created by the workers themselves. There are workers engaged in their own account jobs, while others are involved in private or public institutions in exchange for a fixed or negotiated salary. Small farmers (48.68%) and large farmers (0.66%) are considered to be self-employed, rather than unemployed, in this study. They satisfy the definition of self-job creators as they are performing their own account work (agriculture) for "family gain" compensated for in kind and sometimes in cash. However, their subsistence activities are not considered by the Rwandan Government as providing them with an employed status, according to the revised definition of "self-employment status" [20]. In the current situation of land policy reform, the agricultural population are merely land users. The National Statistics of Rwanda only recognize large farmers who work for profit as self-employees paid in cash.

**Table 3.** Main jobs practiced by the adult members of domestic units.

| Main Jobs Recorded | Percentage of Domestic Unit Members Involved in the Recorded Jobs |
|---|---|
| Small farmer (subsistence agriculture) (The worker is engaged in agricultural production to satisfy the nutritional needs of the family and occasionally sells the surplus.) | 48.68 |
| No job (invalid, housewife, and students) | 13.15 |
| Mining and quarrying (waged job) (Waged job: The worker is paid a discussed salary per day or after the job is finished, but does not have a signed contract (usually oral).) | 0.66 |
| Hair dresser (waged job) | 0.66 |
| Trader (own account job) (Own account job: The worker is engaged in his/her own business, but does not have a certificate of skills or a recognized diploma.) | 8.55 |
| Accounting (salaried profession) (Salaried profession: The worker receives a fixed salary per month and has an official and signed contract.) | 1.32 |
| Construction (waged job) | 9.87 |
| Cloths sewing (own account profession) (Own account profession: The worker is engaged in his/her own business and has a certificate of skills or a valid -diploma.) | 1.97 |
| Director of an International School (salaried profession) | 0.66 |
| Electrician (own account job) | 0.66 |
| Large farmer (own account profession) | 0.66 |
| Teaching (salaried profession) | 1.97 |
| Cow boy (waged job) | 1.97 |
| Nursing (salaried profession) | 0.66 |
| Clothes washing (own account job) | 0.66 |
| Driving (own account and salaried profession) | 4.61 |
| Carpenter (waged job) | 1.97 |
| Security agent (salaried profession) | 0.66 |
| Agricultural laborer (waged job) | 0.66 |
| Total | 100.00 |

Emergent jobs are mainly those in construction (9.87%), trading (8.55%), and driving (4.61%). Car drivers make profits in the transport sector in terms of humans and goods. They are facing competition from motorcycles and bicycles, which are developing a large role in human transport. Other occupations represent less than 30% of domestic unit members involved in the recorded jobs. Hairdressers are paid by the manager of hair salons, but at home they sometimes have their own clients. They have no certificates of competence, but they learn by practice. Traders invest in diversified goods, especially agricultural products and objects of necessity in rural everyday life. Salaried professions with formal contracts in private companies or public institutions are exceptional, representing less than 5% of workers. Clothes washers pass through the villages to offer their services and are paid per quantity and by type of clothes. This job can evolve in a laundry room in a rural environment where electricity is available.

*5.3. Multiple Pathways of Jobs Practiced by Heads of Domestic Units Surveyed in Kimonyi Sector*

Domestic units' heads are engaged in different jobs (Table 4). In total, 47.6% have at least two jobs in their everyday life, 34.9% have only one job and have no other paid occupation, 12.7% have a minimum of three jobs, and 4.8% have no occupation. When someone practices more than one job, they can be considered as being stressed about their economic situation and having an insufficient income from their principal occupation; they are trying to combine many solutions in order to cover their expenses.

**Table 4.** Job diversification by domestic unit head.

| Number of Jobs | Frequency | Percentage |
|---|---|---|
| No job | 3 | 4.8 |
| 1 job | 22 | 34.9 |
| 2 jobs | 30 | 47.6 |
| 3 jobs and above | 8 | 12.7 |
| Total | 63 | 100.0 |

*5.4. Self-Employment Analysis*

Binary logit estimates the factors affecting the decision of the head of the domestic unit to have the principle role of a self-employed worker (Table 5). The results show that the possession of a land certificate, the number of plots (at a 10% significance level), and the ability to read French (at a 1% significance level) positively affect the self-employment resolution for the main activity practiced by the head of the domestic unit. On the contrary, the variables with negative but high and very high significant effects on the decision of the domestic unit's head to practice their main job under a self-employment status include sex (5% significance level), marital status, and reading English (1% significance level). A plausible explanation for this could be that the heads of domestic units are giving priority to stability and full-time salaried work in their first choice of employment to avoid self-employment-associated risks. English-skilled individuals are more likely to be recruited in salaried employment than French-skilled individuals. The latter are trying to find alternative solutions in self-employment opportunities, while the former have more chance of being high cadres in well-structured institutions.

**Table 5.** Factors affecting the decision of the head of a domestic unit to have a principal activity with a self-employment status.

| – | Coef. | St. Err. | $t$-Value | $p$-Value | [95% Conf. | Interval] | Sig. |
|---|---|---|---|---|---|---|---|
| Age | 0.202 | 0.160 | 1.26 | 0.206 | −0.111 | 0.515 | |
| Sex | −17.887 | 7.737 | −2.31 | 0.021 | −33.051 | −2.722 | ** |
| Education | −0.087 | 3.458 | −0.03 | 0.980 | −6.865 | 6.690 | |
| Marital_status | −8.200 | 2.993 | −2.74 | 0.006 | −14.066 | −2.334 | *** |
| Domestic unit's size | −0.503 | 0.525 | −0.96 | 0.338 | −1.533 | 0.526 | |
| Information on expropriation risk | 1.788 | 2.689 | 0.67 | 0.506 | −3.481 | 7.058 | |
| Land Lease Certificate possession | 5.715 | 3.144 | 1.82 | 0.069 | −0.448 | 11.877 | * |
| Number of plots held by the domestic unit | 2.453 | 1.448 | 1.69 | 0.090 | −0.385 | 5.290 | * |
| Total area of land holdings_sqm | 0.000 | 0.001 | −0.47 | 0.635 | −0.002 | 0.001 | |
| Reading Kinyarwanda | −1.301 | 2.807 | −0.46 | 0.643 | −6.803 | 4.201 | |
| Reading French | 14.385 | 4.651 | 3.09 | 0.002 | 5.269 | 23.501 | *** |
| Reading English | −14.693 | 2.850 | −5.16 | 0.000 | −20.279 | −9.108 | *** |
| Reading Swahili | 0.000 | . | . | . | . | . | |
| Constant | 26.264 | 8.728 | 3.01 | 0.003 | 9.157 | 43.371 | *** |

*** $p < 0.01$, ** $p < 0.05$, and * $p < 0.1$.

Binary logit also estimates the factors affecting the decision of the head of the domestic unit to engage in a second activity with a self-employment status (Table 6). The results show that a self-employment decision with regards to a secondary activity is positively and significantly affected by the domestic unit's size at a 10% significance level and reading English at a 1% significance level. A large sized domestic unit has the advantage of labor force availability, while it also requires the earning of more income to satisfy family needs. The variables negatively but significantly affecting the domestic unit's head's involvement in a second activity with self-employment status are the abilities to read French and Swahili (1% significance level). Reading skills in foreign languages are associated with a good level of education. French-skilled individuals are engaging in self-employment for the principal occupation, as discussed in Table 5 above. Therefore, it is very hard to combine two self-

employed positions. However, English-skilled individuals involved in the first activity as self-employed are able to create a second activity on their own account. This result can be explained by disparities in salary. The well-paid employment of English-skilled individuals provides supplementary capital, allowing them to invest in supplementary own account jobs. Swahili is one of the trading languages commonly applied in the East African Community. It is now in the program of school education like other formal languages. Individuals who have good skills in Swahili can have their own capital to create self-employment. They have the advantage of getting paid jobs in the nearest Musanze market. They are also able to profit from the heavy traffic of trans-border businesses in neighboring countries, such as Congo and Uganda.

**Table 6.** Factors affecting the decision of the head of a domestic unit to become involved in a second activity as self-employed.

| Self-Employment_2 | Coef. | St. Err. | *t*-Value | *p*-Value | (95% Conf. | Interval) | Sig. |
|---|---|---|---|---|---|---|---|
| Age | −0.140 | 0.091 | −1.53 | 0.125 | −0.319 | 0.039 | |
| Sex | 5.904 | 3.900 | 1.51 | 0.130 | −1.739 | 13.547 | |
| Education | −2.753 | 1.834 | −1.50 | 0.133 | −6.347 | 0.842 | |
| Marital_status | 2.465 | 1.608 | 1.53 | 0.125 | −0.686 | 5.615 | |
| Domestic unit's size | 0.727 | 0.379 | 1.92 | 0.055 | −0.016 | 1.470 | * |
| Information on expropriation risk | −0.275 | 1.261 | −0.22 | 0.827 | −2.747 | 2.196 | |
| Land Lease Certificate possession | −1.278 | 1.316 | −0.97 | 0.332 | −3.857 | 1.301 | |
| Number of plots held by the domestic unit | −1.002 | 0.761 | −1.32 | 0.188 | −2.494 | 0.489 | |
| Total area of land holdings_sqm | 0.000 | 0.000 | 0.89 | 0.373 | −0.001 | 0.001 | |
| Reading_Kinyarwanda | 0.172 | 1.882 | 0.09 | 0.927 | −3.516 | 3.860 | |
| Reading_French | −29.992 | 5.606 | −5.35 | 0.000 | −40.980 | −19.003 | *** |
| Reading_English | 35.175 | 3.052 | 11.52 | 0.000 | 29.193 | 41.157 | *** |
| Reading_Swahili | −13.998 | 2.688 | −5.21 | 0.000 | −19.265 | −8.730 | *** |
| Constant | −5.614 | 5.212 | −1.08 | 0.281 | −15.830 | 4.602 | |

*** $p < 0.01$, ** $p < 0.05$, and * $p < 0.1$.

### 5.5. Focus Group Synthesis: Qualitative Evidence

The focus group discussion revealed the awareness of the participants regarding improving their self-creativity. The credo known as "*Kwihangira imirimo*" (Promotion of self-created jobs) was recorded in the speech of the focus group participants: "*President Kagame demands every citizen to create his own business*" (our translation). The appeal by the Government to farmers, asking them to be autonomous, independent workers in an effort to work towards self-accomplishment, is translated in Kinyarwanda language as "Twigire muhinzi". Rural women are involved in jobs considered to be masculine, such as the construction of watershed infrastructure, houses, and roads. However, those kinds of jobs are provided by public projects. The popular paradigm of *"kwihangira imirimo"* means the ability to fulfil the condition imposed by the existing labor market, where the services can be accepted and well-paid. The important investment is an open mind for taking any job opportunity, even if it is considered as being only for particular categories, such as men, women, or youth. Participants recognized that autonomy in job creation is limited in terms of one's own capital for investment or the assumption of fiscal responsibilities. They are constrained to maintaining their occupation in subsistence agriculture because of lack of financial and human capital. The dependence on jobs which are conceived by other big investors is a reality in rural areas.

The youth have also been sensitized by the Government to be involved in remunerated jobs to fill the gap between the revenue generated in agriculture and the total expenditures of families. The construction sector is one of the largest providers of manual jobs outside of the agricultural sector.

The trends of professional conversion are also significant in the sector of agri-food technologies. During the "Land Week" (a week of economic activities and public services exposition organized by the District and the Regional Land Center, an occasion for rapidly delivering land certificates to the local population in a short time and through a simplified

procedure) in Rulindo District (1st to 3rd August 2018), in the neighboring District of Musanze, Northern Province, processed food products were exposed for tasting and trading. All of them were made in Rwanda by small and medium enterprises operating in the agro-food sector. Some of these products are produced by social organizations working with marginalized youth. Beneficiaries are engaged in re-education and re-integration, especially for young girls who are victims of sexual abuse. Those single girls usually face discrimination and have a low chance of accessing land through the traditional or reformed land inheritance. The current open land market eases access to land for women and girls. However, land costs are rising and are only affordable for those who have access to a high and guaranteed revenue in the public or private sector.

> *"I'm a nurse and originated of Rulindo District. I come here to make land certificate transfer with the seller originated from Musanze District. I'd like to build my house. My parents gave me a parcel. I got equal size as my brothers. I'll sell a half of my parcel and I'll use complementary incomes with my salary to buy the material for construction"*
>
> (K.M interview, Land Week, Rulindo, 2nd August 2018)

## 6. Discussion

The indicators of literacy status have been analysed due to their significance when the land users need to understand the content of Land Lease Certificates. They are important when someone is looking for another working contract. In particular, the job requirements in the process of recruitment in either the public or private sector are limiting factors for low-skilled candidates. It is also important to be able to communicate and read in different languages, especially those currently used on the local, regional, and international labor market. A high education level has the advantage allowing the practice of foreign languages, needed in professional conversion. The well-educated do not depend on land holdings. In the case of land expropriation, they are able to negotiate fair compensation. Salaried or independent job alternatives are also accessible for them. The 40th indicator of Rwanda Vision 2020 which is related to the literacy rate is projected to be achieved, with a value of 100%, in 2020 [1]. The 2015 Seasonal Agricultural Survey (SAS), season A (from September to February), has reported that 67.1% of agricultural operators attended a primary level of education, 5.2% attended a secondary level of education, 27% have no education, and only 0.8% attended a tertiary level of education [32]. Literacy skills are important for understanding the administrative procedures and "paper-intensive" activity underlying land registration and transactions [33]. A low level of reading skill is disadvantageous for rural people when they are looking for a non-manual job or a profession which requires a high level of education. English and French skills are more important because the highly competitive labor market is structured on a high level of performance in foreign languages. Highly skilled individuals are less likely to be self-employed because they are looking for salaried employment in the public or private sector. Information on expropriation risk is related to job diversification, while it has no effect on self-employment status. Male-headed domestic units have better access to information on land expropriation risk. This result shows that gender is an issue in land use certificate holdings, as has been demonstrated in a Vietnamese study conducted after the "*Doi Moi*" policy in 1986 [11] demonstrating that women are still in a marginal position regarding land holdings [24,34].

The small size of farms observed in Rwanda per household [5,35] enhances the underuse of the labor force. It has been demonstrated that, in Africa, the limits of the capacity to guarantee household needs have already been reached [36]. Food security will be seriously affected by the scarcity of land holdings [15,37]. The minifundial phenomenon has prevailed since the 1980s [28]. Under 0.4 ha per household, it is impossible for landless domestic units to be self-sufficient in terms of food supply and it is not possible only to hang on to subsistence agriculture [36]. When expropriation occurs, small land holders from rural areas struggle to find new plots in surrounding areas. They are obliged to move

far away to the countryside, where job alternatives and self-employment opportunities are exceptional.

Self-employed workers are not working for an employer but are creating a job for themselves or have their own business [38]. A business consists of buying or selling goods and services. In Kinyarwanda, they say "*Umuntu wikorera ku giti cye*" (independent professional) or "*Rwiyemezamirimo*" (entrepreneur) to mean an independent worker who is an investor in her or his own business. They are able to provide employment to other job-seekers. With respect to this definition, farmers could be, for most of the time, "self-employed". However, they are supposed to be unemployed in Rwanda, except farmers who are dealing with agribusiness for market-oriented agriculture [20]. Self-employment is the opposite of "*salaried employment*" [39]. According to the European Labor Force Survey, self-employment refers to people who are engaged in and paid by their own businesses, farming, or qualified professions. On the contrary, people are employed when they get a fixed amount of money paid at the end of a fixed and regular term, usually per month, while they have a formal employment contract.

Peasants are well-known to be able to create their own jobs in farming by replicating the traditional manner of practicing agriculture in their everyday life. However, to create an independent and self-funded job is not as easy as it seems to be. Self-employed people in rural Africa who migrate are mostly those with limited or informal education, while those with a secondary school level education are involved in paid jobs [40]. Expropriation incites double migration from rural areas to cities, and vice-versa. Migrants have the objectives of finding acceptable job alternatives in their new environment. Sedentary domestic units in rural areas also need to enhance their job skills.

Some pertinent criteria which can allow self-employment [41] to be theorized are synthesized in terms of being able to do the following:

- To plan, finance, and execute one's own business;
- To be exposed to financial risk induced by substandard service or the cost of damaged goods;
- To assume enterprise responsibility in investment and management;
- To make profit from rigorous management in efforts to prepare services and goods;
- To control the chain of what is done, the way it is done, and when and how it is done by oneself;
- To be allowed to hire other people to do work defined by oneself in respect of the terms agreed to be assumed;
- To provide authentic services or goods to more than one client and develop several businesses at the same time;
- To make the materials needed available to execute a job;
- To make performant equipment and technology available, rather than poor tools and kits which are not considered as indicators of a specialized business accomplished by a person engaged on their own account;
- To pay taxes and work in conformity with legal business requirements.

According to the criteria described above, the jobs recorded during the field survey in Kimonyi Sector cannot be systematically recognized as self-employment. However, they can be mentioned as job alternatives for avoiding the risk of confusion and misconception regarding the status of self-employment [41]. Turning to non-agricultural activities can be explained by the fact that the capacity thresholds of land holdings are too small to meet the needs of households [36].

Because of the lack of job opportunities, especially for young people, a number of studies have already shown that, in rural areas of Sub-Saharan Africa, there is a massive labor force that is unemployed or underused [42]. Those who are in subsistence agriculture are all considered to be unemployed and constitute a large pool of the labor force, according to the new definition of employment [19,20]. Underemployed are defined as all persons in employment who, during a specified reference period, want to work additional hours, whose working time in all jobs is less than 35 h per week, and who are available to work

additional hours given an opportunity for more work [19,20]. During the field work, it was not possible for us to ascertain how many working hours respondents spent in their on-farming and off-farm activities to evaluate the effective use of their labor force and its remuneration.

## 7. Conclusions and Policy Perspectives

This paper contributes to understanding the situation of self-employment in the context of Rwanda's rural areas affected by land policy reforms, especially the risk of land expropriation due to land use changes. The dynamism of job alternatives is tangible in Rwanda's rural areas. There are diversified jobs in Kimonyi Sector. Agriculture remains the principal occupation in rural areas. The focus of analysis was oriented towards information on imminent expropriation to test the willingness of the population with regards to their professional alternatives. Information on possible risk of expropriation was chosen as a driver of professional shifting. Professional shifting is understood as the possibility of an ever-changing professional profile when the Government decides to change policy, often in unexpected ways. Foreign language skills, such as English, French, and Swahili, need to be improved to enhance the employability of the rural population. The profile of individuals well-informed about land expropriation risk is significantly correlated with English skills, job diversification, male-headed domestic units, and land certificate holdings. Self-employment is negatively influenced by literacy skills in rural areas. The most skilled are attracted by salaried employment.

The credo of self-employment that translates in Kinyarwanda to "*kwihangira imirimo*" in the rural area mind set was approximately interpreted. The focus group debate revealed that it is not understood as self-job creation. It is conceived as willingness towards job changes in a volatile labor market. Participants in the focus group debate assumed that it is best to be prepared for employability in both independent and salaried work. No matter whether a job is created and funded by someone else, the most important aspect is having the skills which match the demand in a highly competitive labor market. The diversified jobs adopted by low-skilled rural people are the most accessible and meaningful off-farm opportunities [43]. However, job alternatives in Kimonyi Sector highlighted land rights uncertainty. Policy makers could base pro-poor planning on the job alternatives identified in rural areas to extend and strengthen self-job creation ability [44]. Workers with a public function may also be given support to plan alternative jobs, such as helping with enhancing their skills in Swahili. The agricultural population could also strengthen their education in foreign languages to enhance their professional mobility and facility to change jobs. In Rwanda, agriculture is gaining appreciation. It is considered a profitable activity not only reserved for low-skilled people. However, investment in agriculture as a business under self-employment status is feasible and interesting for those who have access to a large number of dispersed plots (forthcoming paper). Land market liberalization in rural areas is attracting people who are fascinated by the credo of "self-employment" to invest in large-scale farming. Unfortunately, Rwanda's agrarian system is based on a small land size. For instance, ten ha of land is considered to represent large-scale farming. Land consolidation politics are facing the big issue of economies of scale.

Skills and facilities for self-created employment are limited, but individuals who are well-informed on expropriation risk are more likely to find job alternatives. Therefore, subsequent support is required, especially for the low-skilled population. The booming of the private sector and industrial development can contribute to employment creation. Therefore, the self-created employment program needs to be judiciously implemented across the whole country, both in rural and urban areas, to ensure job accessibility and equity.

The limitation of this study is that the findings are not comparable countrywide due to the small sample and small region covered. Nevertheless, the results constitute a base for reflection of land policy and employment perspectives in rural areas. To avoid the risk of expropriation due to irrational land use [45], it is necessary to optimize self-employment initiatives. Further research is necessary to obtain an in-depth analysis of

the most productive factors and requires the assessment of self-employment promotion in rural areas. The agricultural sector is the transitional rock of self-employment which can create progress in agro-food technologies and social businesses. Furthermore, sporadic and minor jobs created by rural individuals must be recognized in the social mind-set and obtain subsequent institutional support. There should be an exhaustive inventory of all rural economic activities for consideration of equity in terms of remuneration, social protection, and professional development. Therefore, self-employment status needs to be clearly conceived regarding the critical situation of rural people. We recommend further research to ensure more comprehensive and demand-driven self-employment in favor of Rwanda's rural population. Moreover, we recommend a research project monitoring and evaluating the professional conversion of land users expropriated in favor of the "Prime Cement" factory converted in the industrial bloc located in Kimonyi Sector.

**Author Contributions:** M.M. is the corresponding author of this paper. She planned the field work survey and conceived the title. She developed the methodology, wrote and edited the paper. A.M. contributed to statistical analysis and manuscript correction. B.B.M. contributed to manuscript correction. A.K. participated in the research permit facilities, supervised the field work and did comments of the manuscript. P.B. commented on the manuscript and suggested the format of the manuscript and correction. P.L. is the PhD Supervisor. He verified the research protocol before the field work, validated the questionnaire used for survey and the interviews. All authors provided invaluable comments and inputs that helped shape this research, field work, data analysis, and the final version of the manuscript. All authors have read and agreed to the published version of the manuscript.

**Funding:** Field work was supported by the Students Social Services of Gembloux Agro-Bio Tech in 2017. The APC (Article Publication Charges) are provided by the University of Liège. The funders had no role in the design of the study; in the collection, analyses, or interpretation of data; in the writing of the manuscript; or in the decision to publish the results.

**Institutional Review Board Statement:** Not applicable.

**Informed Consent Statement:** Not applicable.

**Data Availability Statement:** The data presented in this study are available on request from the corresponding author (mireille.mizero@doct.uliege.be). The data are not publicly available due to the ongoing processing for a non yet published doctoral thesis.

**Acknowledgments:** The authors wish to thank the Administration of Musanze District and the local administration of Kimonyi Sector for their good collaboration. They facilitate administrative procedures and ease contact with the population during the field work. The authors are also thankful to Ignace Harelimana in charge of the Geographic Information System Data Base at the One Stop Center Office of the Northern Province, Musanze District. He designed the mapping of the study area.

**Conflicts of Interest:** The authors declare no conflict of interest.

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
