# Peer review of "Rwanda’s Land Policy Reform: Self-Employment Perspectives from a Case Study of Kimonyi Sector"

_land, doi:10.3390/land10020117_

Round 1

Reviewer 1 Report

Sorry to say, but I was not impressed by this paper. 

The quality of the English expression is poor, and it was hard to read. 

I failed to grasp the significance of the paper. This was not made clear in the Abstract and Introduction.

The structure of the paper was odd. Normally an introduction does not have subsections or figures.

Reviewer 2 Report

Comments to the authors:

  • It is not clear what are the research objectives of the study and there is no sequence between the objectives and the empirical model specification. The authors should provide in the Introduction the rationale of the topic analysed and then specify it to Rwanda case study. Please don’t forget that the journal targets the international audience and not solely readers from Rwanda. As the paper currently stands, the broader topic analysed is employment diversification in rural areas in the context of land reform policies, which is specified for self-employment for Rwanda; however, the empirical model uses as a dependent variable the risk of land expropriation which completely confuses the reader.
  • Sections 1.1 & 1.2 should form a separate section and not be part of the introduction
  • There is no discussion of the research objectives and the results of the study with findings from the international literature where there are many studies dealing off-farm employment and diversification in rural areas and their determinants.

Reviewer 3 Report

The paper assesses how prepared the rural people of Rwanda are to diversify their jobs in the context of the land policy reform. In all, the topic is also a very relevant one. The comments below can help to improve the article.

The authors seem to make a leap on the subject of expropriation as a central issue in the paper. In the first section, there is one mention of expropriation, and there is nothing on the subject till section 2.4. This leaves a lot of questions unanswered. For instance, what is the nature of the expropriation in Rwanda, and how can it cause the problem the authors are investigating, what kind of compensation is given to the farmers after the expropriation? Monetary or replacement of the land?

Furthermore, what is meant by “expropriation risk”? Is this the risk of the respondents’ being expropriated, or their risk of being destitute and unemployed after the expropriation? This should be explained in the background of the study.

On lines 184 and 186, the authors make mention of non-rational and rational use of the land respectively. What is meant by these terms?

The authors further describe 5 of the 12 variables, including X6,7,8,9,&11, were indicators related to the access to information about expropriation and how to avoid it. However, the paper, as shown in section one, aims at looking at how “Rwanda rural population are prepared to diversify their jobs in the context of land policy reform”. Hence the objective refers to assessing how prepared the farmers are to face the situation after the land reform, and not before and during. This discrepancy should be rectified.

Minor Comments:

The authors should have the paper grammar and spell checked by a native English speaker.

In Figure one, the units on the y-axis should be indicated.

Does the scale on the map really correspond to the distance on the ground? If not, it should be replaced with “not drawn to scale”, or the map should be drawn to scale.

Round 2

Reviewer 1 Report

The paper is a significant improvement, especially in terms of English expression. It is now readable and sufficiently professional. There are still some funny wordings in places, but perhaps this is acceptable. Although I am happy for the paper to be published, I'm not completely convinced of the merit of the paper. The quantitative positivistic approach means that there is little engagement with the real meaning of the issues discussed in the paper.

Reviewer 2 Report

The authors have addressed most of the comments

Reviewer 3 Report

The revision made has taken all the comments made into account.